# Prognostic Value of CAV1 and CAV2 in Head and Neck Squamous Cell Carcinoma

**DOI:** 10.3390/biom13020303

**Published:** 2023-02-06

**Authors:** Jingyu He, Simin Ouyang, Yilong Zhao, Yuqi Liu, Yaolong Liu, Bing Zhou, Qiwen Man, Bing Liu, Tianfu Wu

**Affiliations:** 1Department of Oral and Maxillofacial Surgery, School and Hospital of Stomatology, Wuhan University, Wuhan 430079, China; 2The State Key Laboratory Breeding Base of Basic Science of Stomatology (Hubei-MOST) & Key Laboratory of Oral Biomedicine Ministry of Education, School & Hospital of Stomatology, Wuhan University, Wuhan 430079, China

**Keywords:** bioinformatic analysis, CAV1, CAV2, head and neck squamous cell cancer, immune infiltration, prognosis, biomarker, immunohistochemistry

## Abstract

Background: The CAV family, especially CAV1 and CAV2, is significantly associated with tumor development. In this study, we aimed to explore the pathogenic and prognostic roles of CAV1 and CAV2 in head and neck squamous cell carcinoma (HNSCC) through bioinformatic analysis and verified in human tissue. Methods: We analyzed expression profiles of CAV1 and CAV2 in HNSCC and in normal tissues via data from The Cancer Genome Altas. Prognostic significance was examined by Kaplan–Meier survival curve obtained from the Xena browser together with Cox regression analysis. Co-expressed genes were uploaded to GeneMANIA and applied to Gene Ontology and Kyoto Encyclopedia of Genes and Genomes analyses, showing interaction networks. Signaling pathways of CAV1 and CAV2 in HNSCC were analyzed by Gene Set Enrichment Analysis to elucidate potential regulatory mechanisms. Gene–drug interaction network was explored via Comparative Toxicogenomics Database. Immunohistochemistry was performed to verify theoretical results. Results: Compared with normal tissues, expression levels of CAV1 and CAV2 were remarkably higher in HNSCC (*p* < 0.0001), which independently implies poor OS (CAV1: HR: 1.146, *p* = 0.027; CAV2: HR: 1.408, *p* = 0.002). Co-expressed genes (PXN, ITGA3, TES, and MET) were identified and analyzed by FunRich with CAV1 and CAV2, revealing a significant correlation with focal adhesion (*p* < 0.001), which has a vital influence on cancer progression. GSEA also showed cellular protein catabolic process (ES = 0.42) and proteasome complex (ES = 0.72), which is a key degradation system for proteins involved in oxidatively damaging and cell cycle and transcription, closely correlated with high expression of CAV2 in HNSCC. More importantly, we found the relationship between different immune cell infiltration degrees in the immune micro-environment in HNSCC and expression levels of CAV1/CAV2 (*p* < 0.0001). Gene–drug interaction network was checked via CTD. Moreover, tissue microarrays verified higher expression levels of CAV1/CAV2 in HNSCC (*p* < 0.0001), and the high expression subgroup indicated significantly poorer clinical outcomes (*p* < 0.05). Conclusions: The results revealed that CAV1 and CAV2 are typically upregulated in HNSCC and might predict poor prognosis.

## 1. Background

Head and neck squamous cell carcinoma (HNSCC), accounting for over 90% of head and neck malignant tumor diagnoses, was the dominant form of head and neck cancer and the sixth most frequently diagnosed malignant cancer in the world [1]. In recent years, the survival rate and quality of life for patients with HNSCC are still far from satisfactory, and its incidence is anticipated to keep increasing by 30% by 2030 despite the continuous improvement of diagnostic methods and therapies [2,3]. The occurrence and development of HNSCC is a complex multifactor, multistep and multi-stage process involving a variety of predisposing genetic factors and abnormal expression of genomics and proteomics. Therefore, exploring its specific biomarker and prognostic marker and then discovering new effective options for HNSCC treatment is of vital importance.

The caveolin gene family (CAV1, 2, and 3) is a gene family of cytoplasmic membrane-anchored scaffolding proteins that are widely expressed in most cell types. This gene family encodes proteins caveolins, which are a group of oligomeric structural proteins that are vital for caveolae (Cav) formation [4]. Caveolae are 50–100 nm-shaped invaginations of the plasma membrane that play vital roles as regulators of signal transduction. An increasing number of studies have shown that the CAV family, especially CAV1 and CAV2, is significantly associated with tumor-associated processes. CAV1 and CAV2 regulate processes, including tumor growth, cell migration and metastasis, angiogenesis, and drug resistance [5]. They might be crucial potential molecular targets for cancer treatment. However, whether they function as an oncogene or a tumor suppressor remains controversial and complex [6]. The biphasic functions in HNSCC and some detailed molecular mechanisms of CAV1 and CAV2 still remain undefined.

In this study, we analyzed and compared the expression profiles of CAV1 and CAV2 mRNA in HNSCC and in corresponding normal tissues via the data from The Cancer Genome Atlas (TCGA)-HNSC. We studied their prognostic in HNSC and their potential molecular mechanism and regulatory network. Immunohistochemistry (IHC) was performed to verify the theoretical results.

## 2. Methods

### 2.1. CancerSEA Database Analysis

CancerSEA (http://biocc.hrbmu.edu.cn/CancerSEA/) is the first dedicated database for comprehensively exploring the distinct functional states of cancer cells at the single-cell level. The cancer-related single-cell sequencing datasets for human samples in CancerSEA are derived from 72 datasets in the Sequence Read Archive (https://www.ncbi.nlm.nih.gov/sra), the GEO database, and ArrayExpress (https://www.ebi.ac.uk/arrayexpress/). CancerSEA portrays a single-cell functional cancer state atlas involving 14 functional states of 41,900 single cancer cells from 25 cancer types. It has been reported that CAV family genes are closely associated with tumor-associated processes, but their role in HNSCC and specific mechanisms remain unclear. Therefore, we used the CancerSEA database to verify the correlation between the CAV gene and HNSCC and then analyzed their detailed functional correlation.

### 2.2. Comparison of CAV1 and CAV2 Gene Transcripts in HNSCC

Data in TCGA-HNSC was obtained via the University of California Santa Cruz (UCSC) Xena browser (https://xenabrowser.net/). The patients who had primary tumors and had not received neoadjuvant therapy were included, and patients with missing information were screened out. Their genetic, clinical and survival data were downloaded for analysis.

The alternative transcripts of CAV1 and CAV2 in HNSCC tissues and normal tissues were analyzed. Transcript data in normal tissues were gained from The Genotype-Tissue Expression (GTEx) project, which is an ongoing effort to build a comprehensive public resource to study tissue-specific gene expression and regulation. Gene expression was quantified by RNA-seq (IllumiinaHiSeq). Log2 transcript per million (TPM) was calculated and compared.

### 2.3. Statistics and Survival Analysis

All 510 tumor samples were obtained from the above steps. CAV1 and CAV2 samples were equally divided into 2 groups, which are the high expression group and the low expression group, in accordance with their expression levels. UCSC Xena was used to draw the Kaplan–Meier (K-M) survival curve online, where the parameters are overall survival (OS), disease-specific survival (DSS) and progression-free interval (PFI). When analyzing each survival parameter, the expression of CAV1 and CAV2 was divided into high-expression and low-expression groups according to their median values, and the *p*-value was obtained simultaneously. Based on the known risk factors affecting the survival of HNSCC patients, we conducted a Cox regression analysis. The survival package of R was used for univariate and multivariate Cox regression analysis. For Cox expression analysis, the CAV1 expression and CAV2 expression are continuous variables. All presumed risk factors in the univariate Cox regression analysis were included in the subsequent multivariate Cox regression analysis. IBM SPSS Statistics 21 was used to perform a cross table of different expression groups and different clinicopathological parameters, in which Fisher’s exact test was used to obtain the *p*-value.

### 2.4. Identification of CAV1/CAV2 Co-Expressed Genes

c-CBioPortal for Cancer Genomics (http://www.cbioportal.org/) is a comprehensive open web integrating data mining, data integration and visualization based on the TCGA database developed by Memorial Sloan Kettering Cancer Centre. Cbioportal includes various analysis functions in multi-omics research, including OncoPrint, Cancer Types Summary, Plots, Mutations, Co-expression, Enrichments, Survival, CN Segments, Network and other analysis results. Therefore, the co-expressed genes of CAV1/CAV2 in HNSCC were identified in Head and Neck Squamous Cell Carcinoma (TCGA, Firehose Legacy) (530 total samples) by c-CBioPortal. We identified 20,196 genes in 496 HNSCC and 34 normal tissues using RNA-seq data. The co-expressed genes were identified by the following 2 criteria: (a) The expression of these genes needs to have a strong correlation with CAV1/CAV2 expression (|Pearson’s R| ≥ 0.60) in normal and tumor tissues; (b) if their expression was positively correlated with CAV1/CAV2’s, the expression levels of these genes need to have a significant difference between normal samples and tumor samples (Welch’s *t*-test, *p* < 0.05).

### 2.5. Protein-Protein Interaction (PPI) Network Analysis

GeneMANIA (http://genemania.org/) is a flexible, user-friendly database. It can give hypotheses about gene function by analyzing gene lists and prioritizing genes for functional assays. Given a query gene list, GeneMANIA uses a large number of genomic and proteomic data to find genes with similar functions. It weights each functional genome dataset in accordance with the predicted value of the query. Another use of GeneMANIA is gene function prediction. Given a query gene, GeneMANIA finds the genes that may share functions in accordance with the interaction between them. Thus, we uploaded CAV1/CAV2 and its co-expressed genes to GeneMANIA to analyze the potential interaction networks of their encoded proteins. FunRich (ver.3.13, http://www.funrich.org/) is a stand-alone software tool used mainly for functional enrichment and interaction network analysis of genes and proteins. We submitted CAV1, CAV2 and four co-expression genes into FunRich and analyzed them based on Gene Ontology (GO) database in order to explore the underlying mechanisms and potential impacts of our target genes contributing to HNSCC.

### 2.6. Gene Set Enrichment Analysis (GSEA)

GSEA plays a dominant role in loads of methods and helps explore the mechanisms of the gene that determines whether a predetermined set of genes shows statistical significance between two biological states [7]. In this study, early data processing first generated an ordered list of all genes in accordance with their level of corresponding target genes’ expression levels, and then the relative biological processes or pathways were identified statistically. Gene set permutations were operated 1000 times for every analysis. The expression levels of CAV1 and CAV2 were used as a phenotype label. The nominal *p*-value and normalized enrichment score were used to identify the enriched pathways in each phenotype.

### 2.7. Using TIMER for Tumor-Infiltrating Immune Cell Exploration

The tumor microenvironment determines the progressions of the tumor. We used the online website TIMER (https://cistrome.shinyapps.io/timer/) to analyze the relationship between immune cell infiltration in tumor tissue microenvironments and the expression levels of CAV1 and CAV2. TIMER provides 6 major analytic modules that permit users to multi-directionally explore the connections between immune infiltrates and a wide spectrum of factors, including gene expression, clinical outcomes and somatic mutations [8]. The database contains a variety of tumor-infiltrating immune cells, including B cells, CD4 T cells, CD8 T cells, macrophages, neutrophils and dendritic cells. The purpose of this study is to provide insight into the relationship between the expression levels of target genes and the degree of immune cell infiltration.

### 2.8. CAV-Drug Interaction Network Analysis

To explore the relevance between the CAV and anticancer drugs, we used the data from the Comparative Toxicogenomics Database (CTD, http://ctdbase.org/) to construct a CAV–Drug Interaction Network, which shows the influence of different anticancer drugs on the expression of CAV. We then used Cytoscape (ver.3.7.2) for visualization.

### 2.9. IHC

IHC is an integral technique for tissue-based diagnostics and biomarker detection that is used worldwide and is an extremely valuable supplemental tool to standard morphologic diagnosis in diagnostic pathology. We constructed a tissue microarray (TMA) and collected relevant clinical data for HNSCC patients from the Hospital of Stomatology, Wuhan University (WHUSS), from 2017 to 2022, a total of 172 samples. Data collection and processing were performed in accordance with the World Medical Association Declaration of Helsinki. Sections from each block were cut at 4 μm and stained by using reasonable antibodies for CAV1 and CAV2 proteins collected from Proteintech Company. A total of 172 different samples from the HNSCC patients and their corresponding adjacent normal tissue were stained by using the specific antibodies and were examined by a Scoring System and Analysis Aperio Image Scope CS2 scanner (CA, USA) alongside with Aperio Quantification software (Version 9.1) to quantify the appropriate immunoreactivity of the antibodies towards CAV1 and CAV2 antigens. We then used the formula (1 × the percentage of weakly positive staining) + (2 × the percentage of moderately positive staining) + (3 × the percentage of strongly positive staining) for counting the H-scores of targeting area followed by assessing the H-scores of CAV1 and CAV2 in accordance with the Spearman correlation after shifting to the dataset (Microsoft Excel). We performed survival analysis by using the K–M survival curve to verify the existing statistical result from bioinformatic analysis, which identified the H-score as the measurement tool reflecting the expression levels of CAV1 and CAV2.

## 3. Results

### 3.1. Relevant Clinical Feature

#### 3.1.1. Functions of CAV1 and CAV2 in a Single HNSCC

Heterogeneity related to different functional phenotypes of tumor cells has been a major barrier to the accurate diagnosis and effective treatment of cancer. In recent years, advances in single-cell sequencing (scRNA-seq) technology have provided an opportunity for precisely understanding the functional states of tumor cells at a cellular level. Functional correlation analysis via CancerSEA showed that the functional phenotypes of CAV1 and CAV2 in HNSCs were correlated with HNSC, which included positive correlation with metastasis, invasion, hypoxia, EMT, angiogenesis and differentiation, and negative correlation with stemness. CAV1 showed a moderate correlation with metastasis, hypoxia and invasion (Figure 1a), and CAV2 showed a moderate correlation with metastasis (Figure 1b). CAV1 showed a stronger correlation with HNSCC than CAV2.

#### 3.1.2. CAV1 and CAV2 mRNAs Are Significantly Upregulated in HNSCC Tissues

We made a statistical comparison of the expression levels of CAV1 and CAV2 between 510 cases of HNSCC and 40 cases of adjacent normal tissues via the RNA-seq data in TCGA-HNSC. Analysis and group comparison indicated that the mRNA expression levels of CAV1 and CAV2 genes were significantly higher in tumor tissues compared with those in normal tissues (Figure 2).

#### 3.1.3. CAV1 and CAV2 mRNA Expression Levels Were Associated with Multiple Clinical Features

The CAV1 and CAV2 mRNA expression levels in HNSCC patients with different clinical parameters were significantly associated with the clinical stages of tumors. The gene expression levels were significantly lower in patients with advanced clinical stages (III/IV vs. I/II) and advanced nodal invasion stages. These outcomes all manifested that the expression levels of CAV1 and CAV2 were remarkably correlated with different clinical stages (Figure 3).

#### 3.1.4. Statistics and Survival Analysis

The baseline information of all 510 samples obtained is shown in Table 1. Fisher’s exact test was conducted, showing that between the CAV1 high expression group and low expression group, significant differences were found in radiation therapy (*p*-value = 0.038). Significant differences were observed in gender and clinical stage between the CAV2 high expression group and the low expression group (*p*-value = 0.020, 0.019). In the K-M survival analysis, CAV1 and CAV2 were divided into the high-expression group and the low-expression group by median values. K–M survival analysis showed that higher CAV1 and CAV2 expression was significantly associated with worse PFI, DSS and OS at both the 3-year cutoff and 5-year cutoff. (Figure 4 and Appendix A) Cox regression analysis showed that CAV1 expression may be a prognostic factor of DSS and OS, and CAV2 expression may be an independent prognostic factor of DSS, OS and PFI (Table 2).

Table 2 The result of univariate and multivariate Cox regression analysis indicates that CAV1 expression may be a prognostic factor of DSS and OS, and CAV2 expression may be an independent prognostic factor of DSS, OS and PFI. All *p*-values less than 0.05 are bold.

#### 3.1.5. CAV1 and CAV2 mRNA Expression Levels Were Correlated with Primary Tumor Sites

We performed data analysis between the expression levels of CAV1 and CAV2 with the location of the primary tumor and patients’ HPV status. HNSCC contains tumors of multiple sites, including tonsils, hypopharynx, palate, oropharynx, etc. The expression of CAV1 was lowest in tonsil cancer and highest in palate cancer, and the expression of CAV2 was lowest in tonsil cancer and highest in hypopharynx cancer. Compared with other primary tumor sites in HNSCC, the sites with the highest and lowest expression of target genes have significant differences (Figure 5).

#### 3.1.6. CAV1 and CAV2 mRNA Expression Levels Were Associated with HPV Status

Human papillomavirus (HPV) infection and p16 patterns are vital prognostic factors for certain Head and Neck malignant tumors, especially for HNSCC. RNA-seq data in TCGA-HNSC on the HPV status of patients by p16-testing was collected and analyzed. The results showed that the mRNA expression levels of CAV1 and CAV2 genes were significantly higher in HPV-negative samples compared with those HPV-positive samples (Figure 6). According to our analysis, higher expression of CAV genes is related to poorer outcomes, which is consistent with previous reports of improved outcomes in HPV-positive HNSCC patients [9]. Compared to HPV-negative tumors, HPV-positive tumors are genetically different and inversely correlated with biomarkers for poor prognosis (e.g., p53 mutations) [10,11], which determines the molecular profile of tumors and thus the tumor-associated process (e.g., infiltrates of immune cells [12]) These might explain the higher rates of response to radiation therapy and chemotherapy and better in HPV-positive tumors. The differentially expressed CAV genes might play a role in these differences in molecular profiles and, as a consequence, lead to poor prognosis outcomes.

### 3.2. Molecular Interaction

#### 3.2.1. PPI Co-Expression

To further identify the potential mechanism network of CAV1/CAV2 in HNSCC, we identified 20,196 genes with RNA-seq data in 496 HNSCC and 34 normal tissues. The co-expressed genes were identified by two criteria as follows: (a) The expression of these genes needs to have a strong correlation with CAV1/CAV2 expression (|Pearson’s *R*| ≥ 0.60) in normal and tumor tissues; (b) if their expression was positively correlated with CAV1/CAV2′s, the expression levels of these genes need to have a significant difference between normal samples and tumor samples (Welch’s *t*-test, *p* < 0.05). In accordance with the above two criteria, we identified four co-expressed genes, namely, PXN, ITGA3, TES and MET. All these genes showed positive correlations with CAV1/CAV2 expression (Figure 7a). Subsequently, we uploaded these co-expression genes together with CAV1/CAV2 to GeneMANIA to analyze their internal protein interaction network. The results showed that many close molecular interactions were identified among the proteins encoded by these genes. We found that 36.96% had similar expression characteristics, 31.90% had shared protein domains, 10.51% had physical interactions, and 6.03% exerted colocalization. All results, including the predicted pathway, are shown in Figure 7b.

Moreover, the enrichment analysis of CAV1 and CAV2 and co-expressed genes using FunRich based on the GO database showed the significant enrichment of focal adhesions in the aspect of cellular component (Bonferroni method *p*-value < 0.001, Figure 7c). It suggested that the gene products of target genes may be positioned at focal adhesion, which has already been proven to play an important role in the adhesion and metastasis of tumor cells [13].

#### 3.2.2. GSEA

GSEA was performed to identify the GO terms and signaling pathways in the low and high CAV1 and CAV2 expression groups of patients with HNSCC based on the TCGA database. We found that the cellular protein catabolic process (ES = 0.42) and proteasome complex (ES = 0.72) were enriched in the CAV2 high expression subgroup (Figure 8). The proteasome is the key degradation component for oxidatively damaged proteins, as well as multiple proteins involved in the cell cycle and transcription, both of which are vital for cancer development [14,15], thereby supporting the significant correlation between the high expression of target genes and the possibility of the occurrence of HNSCC.

#### 3.2.3. Using TIMER for Tumor-Infiltrating Immune Cell Exploration

We found that the higher expression of CAV1 was negatively correlated with T cell regulatory (Tregs) (partial.cor = −0.341 *p* = 6.82 × 10^−15^) and positively correlated with CD8 + T cells (partial.cor = 0.383 *p* = 1.36 × 10^−18^), CD4 + T cells (partial.cor = 0.285 *p* = 1.19 × 10^−10^) and B cells (partial.cor = 0.164 *p* = 2.65 × 10^−4^) via searching the relationship between the expression levels of CAV1 and CAV2 and the infiltration degree of different types of cells in immune microenvironment in HNSCC. The higher expression of CAV2 was negatively correlated with the infiltration of B cells (partial.cor = −0.257 *p* = 7.51 × 10^−9^) and Tregs (partial.cor = −0.275 *p* = 5.25 × 10^−10^) but positively correlated with the degree of immune infiltration of CD4 + T cells (partial.cor = 0.402 *p* = 1.53 × 10^−20^) and CD8 + T cells (partial.cor = 0.205 *p* = 4.78 × 10^−6^; Figure 9).

#### 3.2.4. CAV-Drug Interaction Network Analysis

We constructed the CAV–drug interaction network by using the data from CTD. Thirteen anticancer drugs can influence the expression levels of CAV1 and CAV2. As shown in the figure, 11 drugs can influence the expression level of CAV1, four drugs can influence the expression level of CAV2, and two drugs can influence the expression levels of CAV1 and CAV2 (Figure 10).

### 3.3. Experimental Validation

After performing IHC in 172 samples, we found that CAV2 and CAV1 were highly expressed in HNSCC tissues. To validate the expression levels of CAV1 and CAV2 in HNSCC, we further analyzed the different expression levels in HNSCC tumors and adjacent normal tissues. The expression levels of CAV1 and CAV2 in HNSCC tissues were significantly upregulated compared with the normal tissues, as shown in Figure 11a. This process was also performed to evaluate the association of CAV1 and CAV2 expression levels in the adjacent normal tissues and HNSCC tissues of 129 cases for CAV1 and 143 cases for CAV2. CAV1 included 34 patients with normal oral mucosa (OM) and 95 patients with HNSCC, and CAV2 included 34 patients with OM and 109 patients with HNSCC. The IHC staining demonstrated an accurate H-score for CAV1 and CAV2 in these tissues. The significant ascent was observed compared with the H-score in HNSCC with that in OM (Figure 11b), further proving the positive correlation between the expression levels of CAV1 and CAV2 in HNSCC with the OM. We ascertained the previous statistical conclusion that the low expression levels of CAV1 and CAV2 suggest superior prognostic performance in HNSCC patients through survival analysis targeting our available samples (Figure 11c).

## 4. Discussion

HNSCCs are the most common malignant tumor that occurs in the head and neck [2]. As the sixth most common cancer worldwide, with 890,000 fresh cases and 450,000 deaths in 2018, the occurrence of HNSCC continues to increase and is forecasted to reach 1.08 million annually by 2030 [16]. Scientists are continuously searching for possible solutions against HNSCC, including immunotherapy that is strongly related to different immune cells consisting of tumor microenvironments, such as PD1/PDL1 [17], and regular medical treatment with chemotherapy. A significant biomarker that involves the crucial signal pathways during the process of the HNSCCs’incident development, together with prognosis and some specific evaluation criteria such as the REASON score, a promising biomarker to predict the risk of mortality in early-stage HNSCC patients, is helpful to provide a method for precisely cutting off some particular ways that promote HNSCC [18]. To search for a feasible biomarker for HNSCC, we made a series of explorations by using bioinformatic methods and further experiments, such as IHC, to verify the final outcome.

CAV1 and CAV2 are the two main members of the CAV gene family, which are in charge of encoding the caveolins, the essential component of the plasma membrane. Our results show that CAV1 and CAV2 exhibit obviously higher expression levels in HNSCC than that in adjacent normal tissue with a worse prognosis. Previous studies suggested that the expression of CAV1 might act as a bidirectional factor for many malignancies, such as lung, breast and pancreatic cancers. The objectionable effects on tumor progression and prognosis are reasonable due to their functions for promoting cancer invasion, regulating metabolism and restraining different drugs [19,20,21]. However, some studies have insisted that CAV-1 might function as a tumor inhibitor in diverse types of cancer, such as lung, breast and pancreatic cancers. Few studies have reported that CAV2 exhibits different effects in breast, oesophageal and pancreatic cancers [22,23,24,25]. Despite some of the studies on CAV1 and CAV2 in HNSCC have collected basic information, such as gene expression and analysis of survival, to prove the potential mechanism underlying the promotional effect and possibly the inhibitory effect in HNSCC progression [6], the accurate role that CAV1 and CAV2 play in HNSCC is still indistinct. Thus, we conducted a systematical work of bioinformation-related data collection and verification with experiments from various aspects. A series of methods for gathering available information related to CAV1 and CAV2 was used to determine the possible reason behind the higher expression levels of our target genes in HNSCC. However, some inevitable limitations still existed in our study, which are attributed to the small number of confirmatory experiments and inadequate sample capacity. We solved the problem by simply increasing the amount of patients sample and using IHC to fill the void in the experiments.

The late 20th century witnessed the growth of bioinformatic methods applied to traditional biology [26]; their use, along with the continuously improving laboratory technology, has given a chance to solve complicated questions by reaching a completely fresh stage of information retrieval and data analysis that conclude the mechanism behind the tumor incidence, progression and prognosis. Determining the accurate interactions with particular genes or proteins that manifest extraordinary expression is more efficient, thereby allowing more efficient therapeutic treatment to improve the possibilities for the cure of the disease. We use bioinformatic means as our crux methods, including three phases directly toward the final conclusions. In Phase 1, we downloaded the original data from the UCSC Xena browser (https://xenabrowser.net/) and used the GraphPad prism for the visualization, indicating that the variations in CAV1 and CAV2’s expression levels are related to HNSCC. (1) Tumour tissues share significantly higher expression levels of CAV1 and CAV2 compared with the adjacent normal tissues. (2) HPV-positive patients have significantly lower expression of CAV1 and CAV2. (3) Different expression levels between HNSCC at different sites indicate that our target genes have potential possibilities for being independent biomarkers in HNSCC. In Phase 2, we performed a whole train of work from different perspectives for deeper analysis in terms of various features of our target genes, from genes’ expression to the interaction between the related protein and the corresponding drugs for therapy. K–M survival analysis illustrated that the CAV1 and CAV2’s higher expression subgroup has a worse prognosis than the lower expression subgroup. The univariate analysis indicated that high CAV expression was associated with poorer OS, DSS and PFI. Other clinicopathological parameters, including age, gender pathological stage, T stage, N stage and M stage, were correlated with the prognosis of patients with HNSCC via Cox regression analysis, indicating that CAV1 expression may be a prognostic factor of DSS and OS, and CAV2 expression may be an independent prognostic factor of DSS, OS and PFI. To determine the underlying mechanism of CAV1 and CAV2 in HNSCC, we used the online website database Xena with two of the main criteria to screen the appropriate genes that have a strong correlation with the expression levels of CAV1, and CAV2 (|Pearson’s *R*| ≥ 0.60) and are significantly relevant (Welch’s *t*-test, *p* < 0.05). In accordance with the above two criteria, four genes with high connectivity were filtered from this module. They are PXN, ITGA3, TES and MET, which have a positive relationship with the expression levels of CAV1 and CAV2. We then visualized these genes on Xena, showing that their higher expression occurs in the tumor tissues. For the level of proteins, we uploaded the data to the GeneMANIA to validate the four co-expressed genes and applied GO and KEGG analyses to analyze their possible interaction network. The results showed that large proportions of the proteins expressed by the related genes share a similar manner of expression (36.96%) or have particularly the same protein domains (31.90%). Furthermore, through FunRich enrichment analysis, CAV1, CAV2 and four co-expressed genes were found to be significantly correlated with focal adhesion. Focal adhesion is a cellular component made up of a wide range of prosurvival signaling molecules, such as integrins, growth factor receptors, and intracellular molecules, which influence cell activity and have an impact on tumor cell survival and might be used as cancer targets [13]. Therefore it might be a potential mechanism of CAV1 and CAV2 influencing the development of HNSCC. Next, the signaling pathways of CAV1 and CAV2 in HNSCC were analyzed by GSEA. The results show that two signaling pathways, cellular protein catabolic process (ES = 0.42) and proteasome complex (ES = 0.72), which is the primary degrading mechanism for oxidatively damaged proteins as well as a number of proteins involved in the cell cycle and transcription, both of which are crucial for cancer development, obtaining strong correlation with the high expression of CAV2 in HNSCC patients, indicating their significant relation with HNSCC [14]. In Phase 3, immunohistochemical analysis was performed, and the results showed that the expression levels of CAV1 and CAV2 were higher in HNSCC tissues than in normal tissues. We used the H-score to statistically evaluate the exact connection between the expression levels of CAV1 and CAV2 in the OM and in HNSCC, equally leading to the same result that the expression levels of target genes increased in HNSCC. The K-M survival curve further proved our previous conjecture that CAV1 and CAV2 could be the possible prognostic biomarkers in HNSCC, showing the significant connection between the low expression levels of target genes and positive prognostic results.

In summary, we applied several bioinformatic methods to explore the latest datasets and visualize them for intuitive observation. Our study suggested that CAV1 and CAV2 possess higher expression levels in HNSCC than in normal tissues, and the increase in expression represented poorer prognosis in HNSCC patients with positive, thereby verifying the experiments. Our target genes show a significant correlation in different aspects of HNSCC biologically and statistically, thereby providing a suitable direction for therapy against HNSCC. The precise mechanism behind the data needs to be clear, and the accurate approaches for elucidating the phenomenon must be further investigated.

## 5. Conclusions

Our study using bioinformatic approaches indicate the strong relationship between the ascent of the expression levels of CAV1 and CAV2, and HNSCC, which are elucidated statistically and experimentally. The potential methods that can provide different possibilities for therapy against HNSCC. The underlying mechanism still needs further discussion.

## Figures and Tables

**Figure 1 biomolecules-13-00303-f001:**
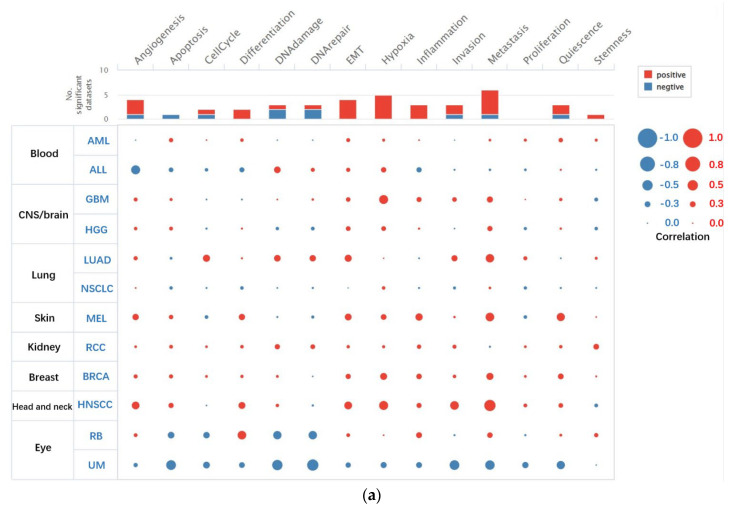
Functions of CAV1 and CAV2 in a single HNSC. Functional correlation analysis via CancerSEA shows that the functional phenotypes of CAV1 (**a**) and CAV2 (**b**) are moderately correlated with HNSCC.

**Figure 2 biomolecules-13-00303-f002:**
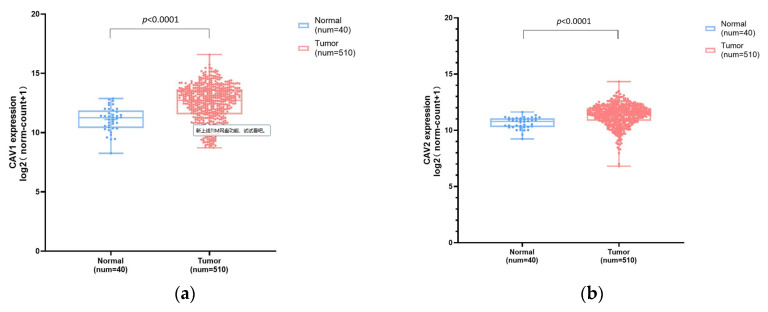
Comparison of mRNA expression levels of CAV1 and CAV2 between HNSC and normal tissues. Box plot chart ((**a**), mean ± SD: Normal: 11.11 ± 1.02, Tumor: 12.51 ± 1.53; (**b**), mean ± SD: Normal: 10.65 ± 0.50, Tumor: 11.35 ± 0.95) showing the expression levels of CAV1 and CAV2 in HNSC and their adjacent normal tissues.

**Figure 3 biomolecules-13-00303-f003:**
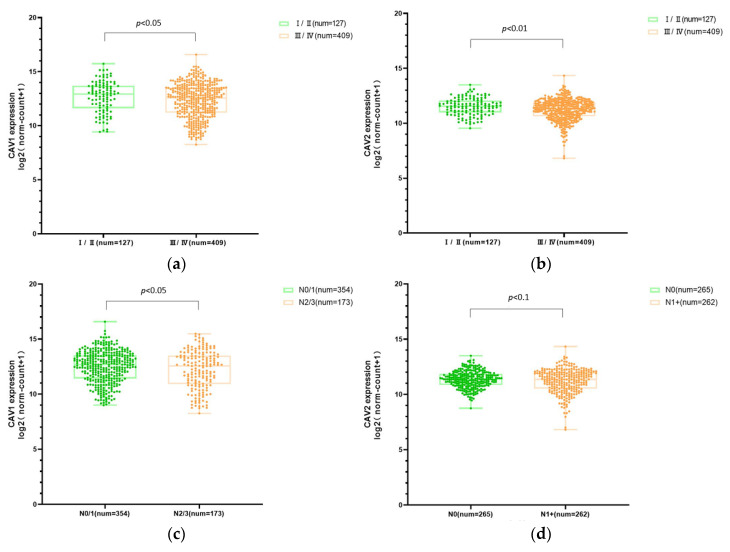
CAV1 and CAV2 mRNA expression levels are associated with multiple clinical pathological features. The box plot chart shows the comparison of mRNA expression levels of CAV1 and CAV2 in HNSCC cases grouped in accordance with clinical stages ((**a**), mean ± SD: Ⅰ/Ⅱ: 12.66 ± 1.365, Ⅲ/Ⅳ: 12.32 ± 1.598; (**b**), mean ± SD: Ⅰ/Ⅱ: 11.50 ± 0.783, Ⅲ/Ⅳ: 11.22 ± 0.983) and nodal invasion status ((**c**), mean ± SD: N0/1:12.49 ± 1.448, N2/3: 12.19 ± 1.723; (**d**), mean ± SD: N0: 11.36 ± 0.765, N1+: 11.21 ± 1.095). N0, nodal negative; N1+, N1/2/3 cases.

**Figure 4 biomolecules-13-00303-f004:**
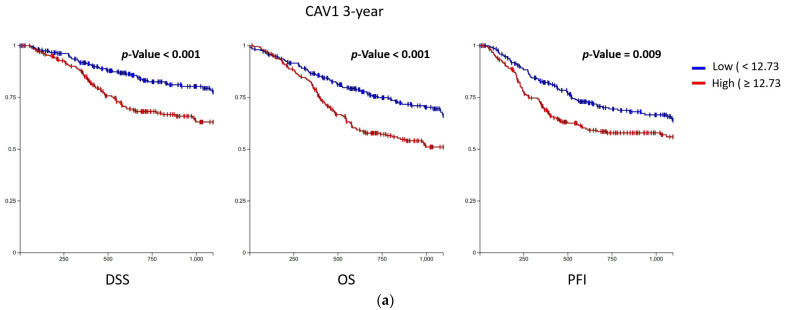
K–M survival curve. The DSS, OS and PFI of CAV1 (**a**) and CAV2 (**b**) are shown at a 3-year cutoff, respectively. Significant differences are found in multiple survival parameters between the high-expression group and the low-expression group.

**Figure 5 biomolecules-13-00303-f005:**
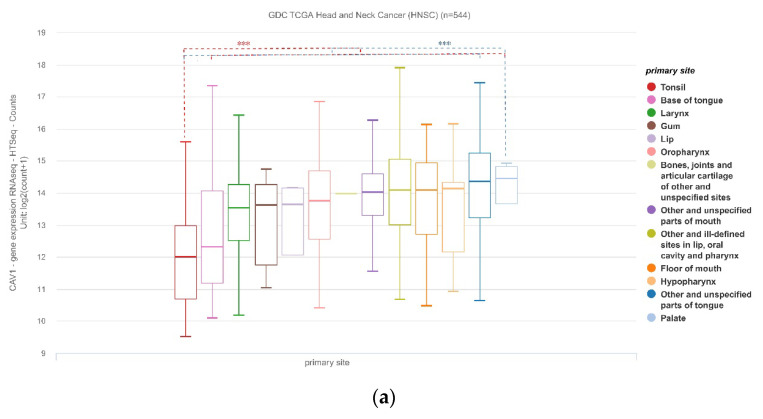
HNSCC contains tumors from multiple sites, including tonsil, hypopharyngeal, palate and oropharynx, etc. (**a**) Expression of CAV1 at different sites, with the lowest in tonsil and highest in the palate (*p* < 0.001); (**b**) Expression of CAV2 at different sites, with the lowest in tonsil and highest in the hypopharynx (*p* < 0.001).

**Figure 6 biomolecules-13-00303-f006:**
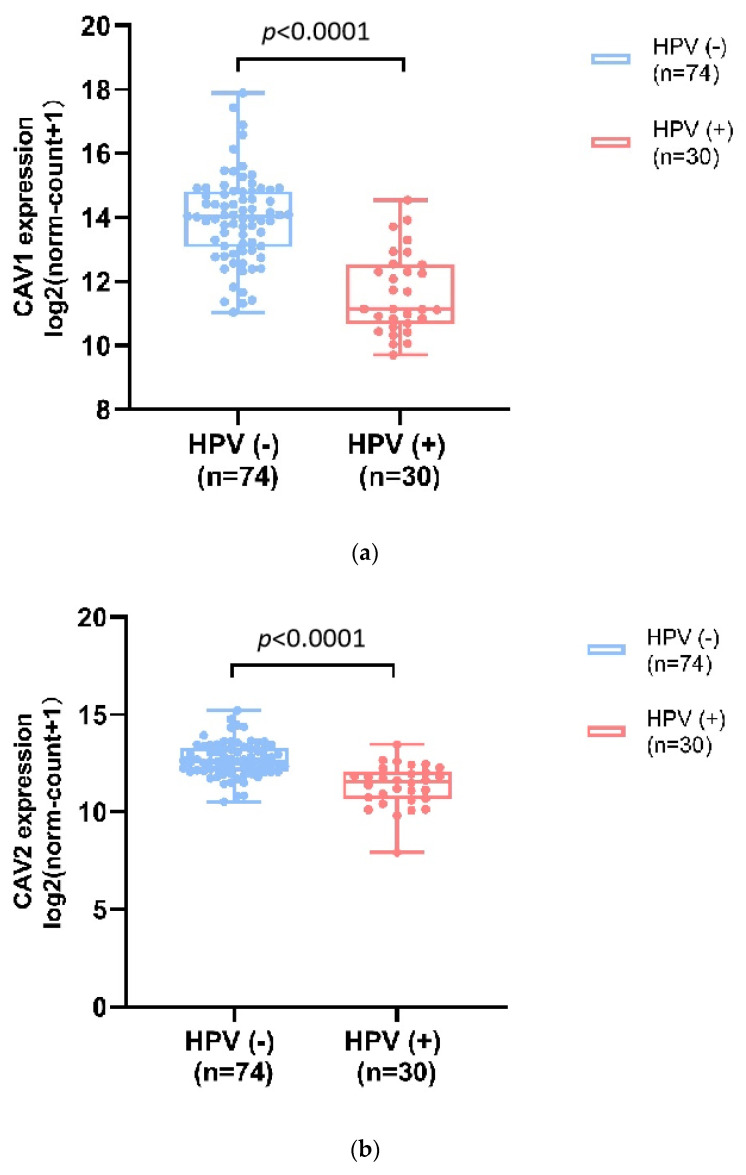
CAV1 and CAV2 mRNA expression levels were associated with HPV status. CAV1 ((**a**), mean ± SD: HPV−:13.97 ± 1.36, HPV+:11.63 ± 1.26) and CAV2 ((**b**), mean ± SD: HPV−:12.69 ± 0.88, HPV+: 11.33 ± 1.09) expression levels were significantly higher in HPV negative samples compared with those HPV positive samples RNA-seq data in TCGA-HNSC on the HPV status of patients by p16-testing was collected and analyzed by Welch’s *t*-test.

**Figure 7 biomolecules-13-00303-f007:**
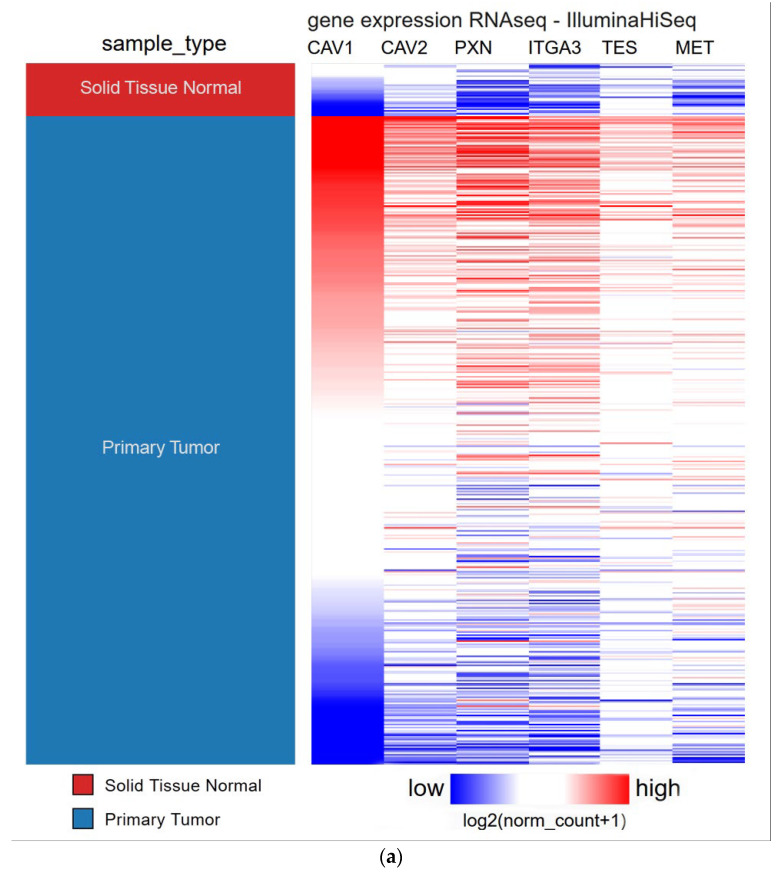
Potential regulatory network of CAV1 and CAV2 in HNSCC (**a**). Heatmap showing the expression profile of four high-potential candidate genes. Interconnections between proteins are explored in terms of co-expression, shared protein domains, predicted physical interactions, colocalization and pathway (**b**). In the GO domain of cellular component, significant enrichment was found in focal adhesion (Bonferroni method *p*-value < 0.001), which demonstrates that the gene products of CAV1 and CAV2 and co-expression genes may be located at focal adhesion (**c**).

**Figure 8 biomolecules-13-00303-f008:**
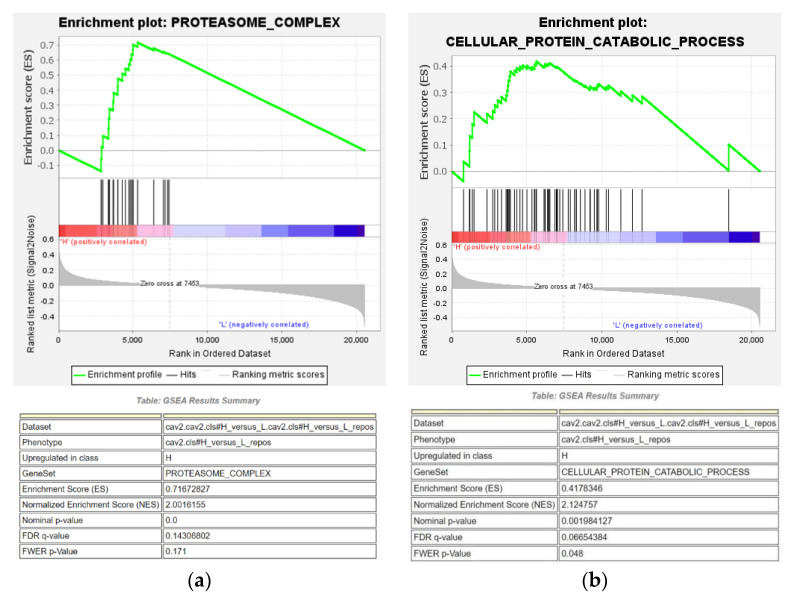
Cellular protein catabolic process (**a**) and proteasome complex (**b**) are significantly enriched in the CAV2 high expression subgroup (*p*-value < 0.02, FDR *q*-value < 25%), thereby providing a feasible hypothesis for particular signal pathways involved in our target genes.

**Figure 9 biomolecules-13-00303-f009:**
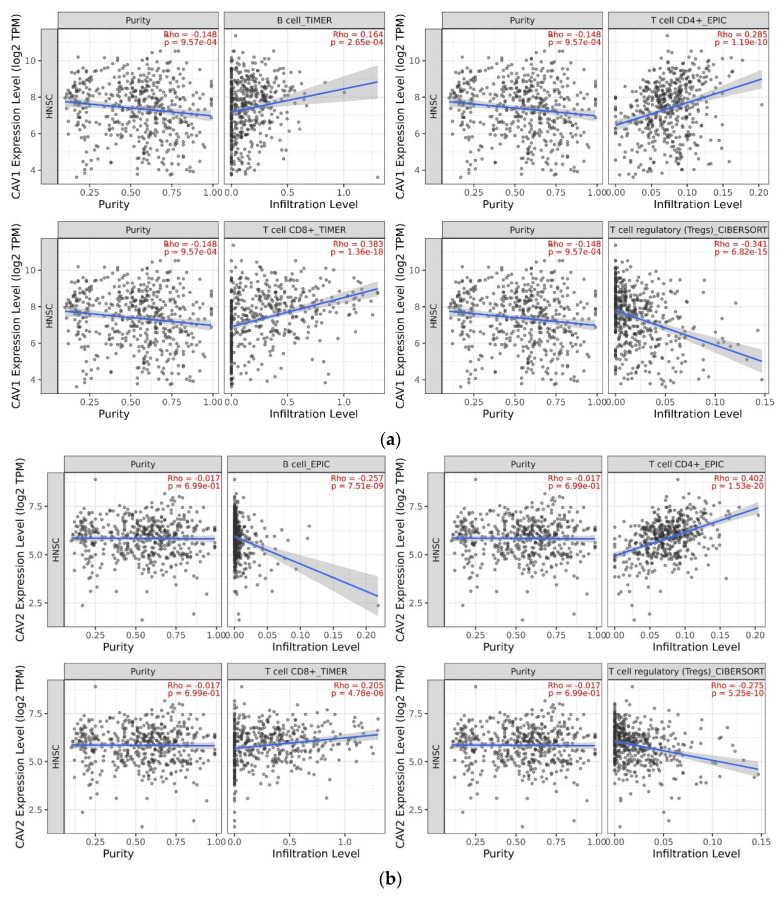
Higher expression of CAV1 is related to the downregulation of Tregs cells but shows the opposite result in CD8 + T cell, CD4 + T cell, and B cell. (**a**) Higher expression of CAV2 is related to the decrease in immune infiltration of Tregs cell and B cell and demonstrates an upregulation effect on CD8 + T cell and CD4 + T cell (**b**) in the microenvironment.

**Figure 10 biomolecules-13-00303-f010:**
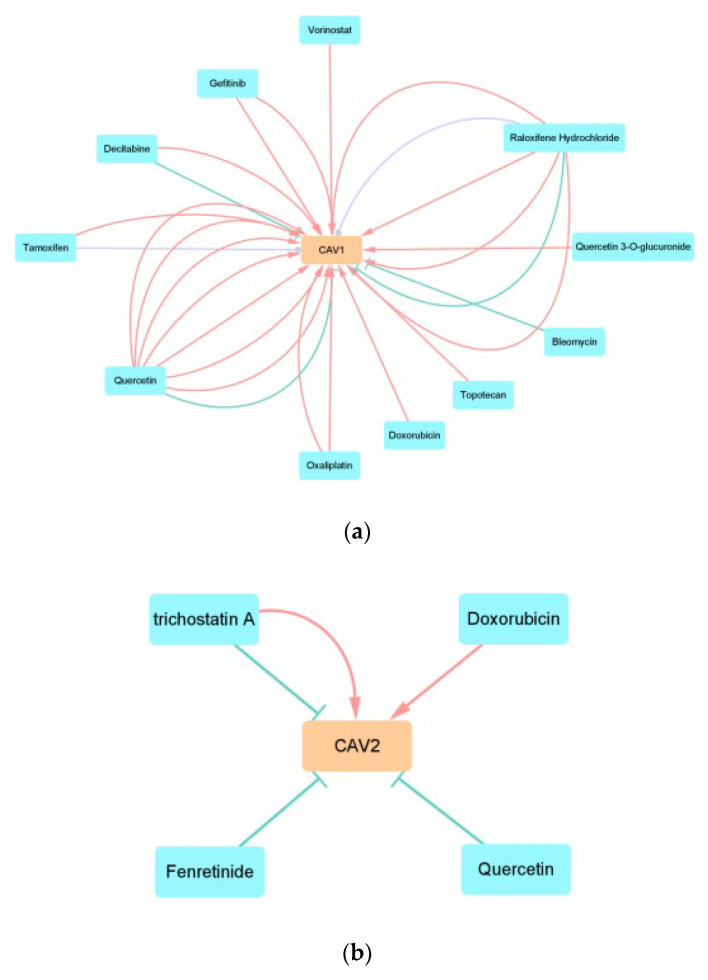
We construct the CAV–drug interaction network with the two genes and anticancer drugs. (**a**,**b**) *T* represents that the drug can decrease the expression of the gene, delta represents that the drug can increase the expression of the gene, and the circle represents that the drug can affect the expression of the gene. The number of lines represents the number of previous studies.

**Figure 11 biomolecules-13-00303-f011:**
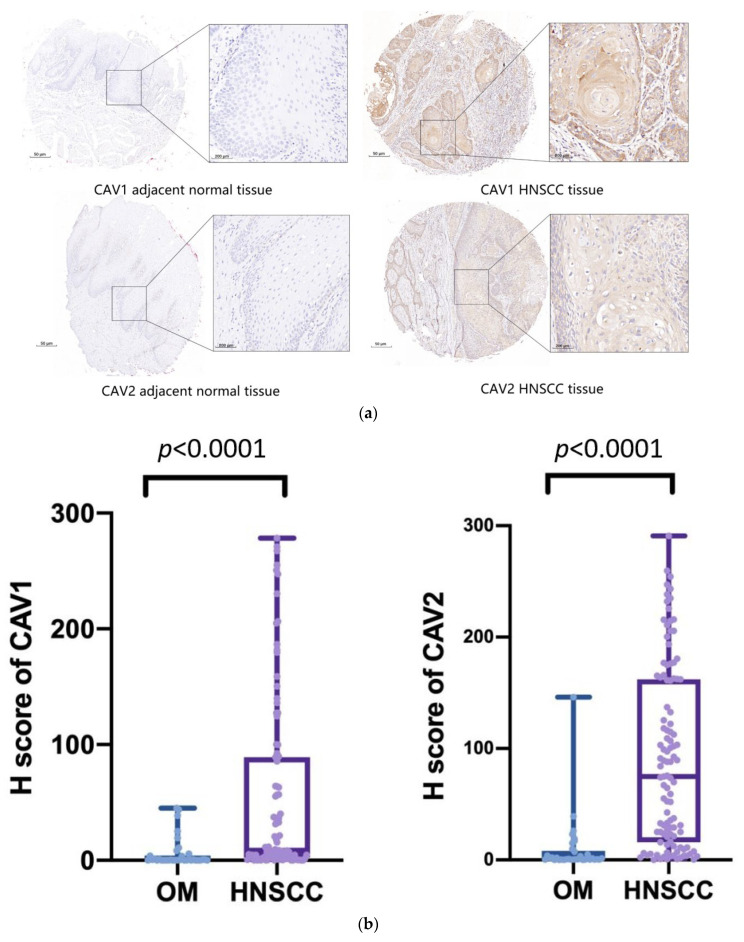
IHC analysis of cancer and paracancerous tissues in 172 patients confirms the CAV1 and CAV2 protein levels in HNSCC tissues, revealing that CAV1 and CAV2 are upregulated in HNSCC (**a**). The result shows a significant correlation is found for H-score in HNSCC and in OM cases after comparing the H-score, which represents the expression levels of target genes (*p* < 0.0001) ((**b**); mean ± SD: 6.21 ± 0.07; mean ± SD: 54.89 ± 2.36; mean ± SD: 9.72 ± 0.07; mean ± SD: 90.64 ± 95.81). Survival analysis shows a significant correlation between the low expression levels of CAV1 and CAV2 and longer survival days, indicating better prognostic performance (**c**).

**Table 1 biomolecules-13-00303-t001:** Baseline information of the 510 patient samples.

		CAV1 Expression			CAV2 Expression		
Parameters		High(N = 255)	Low(N = 255)	*p*-Value	High(N = 258)	Low(N = 252)	*p*-Value
Age	Mean ± SD	61.19 ± 12.452	60.88 ± 11.091	0.769	61.67 ± 12.394	60.38 ± 11.098	0.218
	No data	1	0		0	1	
Gender	Female	76	57	0.069	79	54	0.020
	Male	179	198		179	198	
Alcohol history documented	No	75	82	0.563	82	75	0.773
	Yes	175	168		173	170	
	No data	5	5		3	7	
Tobacco smoking history	No	62	53	0.395	61	54	0.526
	Yes	187	196		190	193	
	No data	6	6		7	5	
Radiation therapy	No	91	65	0.038	86	70	0.165
	Yes	139	151		139	151	
	No data	25	39		33	31	
Neoplasm histologic grade	G1/G2	193	169	0.104	186	176	1.000
	G3/G4	59	73		68	64	
	No data	3	13		4	12	
Clinical T	T1/T2	89	92	0.852	90	91	0.780
	T3/T4	158	155		160	153	
	No data	8	8		8	8	
Clinical N	N0	120	117	0.717	120	117	0.857
	N1/N2/N3	122	129		125	126	
	No data	13	9		13	9	
Clinical stage	Stage I/Stage II	65	48	0.069	68	45	0.019
	Stage III/Stage IV	182	201		182	201	
	No data	8	6		8	6	
Pathological stage	Stage I/Stage II	58	40	0.169	57	41	0.421
	Stage III/Stage IV	174	167		181	160	
	No data	23	48		20	51	

The baseline information of the 510 patient samples obtained from TCGA-HNSC.

**Table 2 biomolecules-13-00303-t002:** Cox regression analysis.

	Univariate Analysis				Multivariate Analysis			
	HR	HR.95L	HR.95H	*p*-Value	HR	HR.95L	HR.95H	*p*-Value
DSS								
CAV1	1.156	1.026	1.303	**0.017**	0.849	0.629	1.147	0.287
CAV2	1.321	1.076	1.620	**0.008**	1.891	1.145	3.124	**0.013**
Age	1.010	0.994	1.026	0.238	1.022	0.998	1.045	0.068
Gender	1.057	0.707	1.579	0.788	0.847	0.475	1.512	0.575
Alcohol history documented	1.216	0.816	1.811	0.337	1.327	0.774	2.275	0.304
Tobacco smoking history	1.085	0.708	1.664	0.707	0.776	0.437	1.375	0.385
Radiation therapy	0.783	0.513	1.194	0.256	0.518	0.288	0.932	**0.028**
Neoplasm histologic grade	1.043	0.806	1.349	0.751	1.166	0.795	1.710	0.431
Clinical T	1.103	0.918	1.326	0.296	1.096	0.711	1.689	0.677
Clinical N	1.258	1.042	1.520	**0.017**	1.494	1.051	2.125	**0.025**
Clinical stage	1.071	0.879	1.305	0.497	0.622	0.349	1.110	0.108
Pathologic stage	1.550	1.210	1.987	**0.001**	2.535	1.620	3.966	**0.000**
Tumor site (sorted by CAV1 expression level)	1.039	0.992	1.089	0.101	0.995	0.849	1.165	0.946
Tumor site (sorted by CAV2 expression level)	1.050	0.997	1.105	0.063	1.054	0.890	1.249	0.539
OS								
CAV1	1.129	1.029	1.238	**0.010**	0.881	0.700	1.108	0.280
CAV2	1.252	1.069	1.465	**0.005**	1.617	1.089	2.400	**0.017**
Age	1.025	1.012	1.038	**0.000**	1.026	1.008	1.045	**0.004**
Gender	1.332	0.994	1.786	0.055	0.964	0.624	1.491	0.870
Alcohol history documented	0.934	0.699	1.249	0.646	1.033	0.696	1.533	0.873
Tobacco smoking history	1.141	0.809	1.609	0.453	0.894	0.570	1.402	0.626
Radiation therapy	0.609	0.446	0.831	**0.002**	0.443	0.288	0.681	**0.000**
Neoplasm histologic grade	1.012	0.831	1.233	0.903	1.096	0.829	1.449	0.519
Clinical T	1.081	0.937	1.247	0.288	0.950	0.684	1.320	0.761
Clinical N	1.141	0.983	1.324	0.084	1.144	0.875	1.497	0.326
Clinical stage	1.084	0.929	1.264	0.307	0.884	0.569	1.374	0.584
Pathologic stage	1.438	1.200	1.723	**0.000**	2.415	1.724	3.384	**0.000**
Tumor site (sorted by CAV1 expression level)	1.028	0.991	1.066	0.134	1.072	0.920	1.248	0.373
Tumor site (sorted by CAV2 expression level)	1.028	0.988	1.070	0.171	0.953	0.806	1.126	0.569
PFI								
CAV1	1.094	0.994	1.204	0.065	0.801	0.637	1.007	0.057
CAV2	1.233	1.046	1.454	**0.013**	1.793	1.212	2.652	**0.003**
Age	1.007	0.994	1.020	0.309	1.014	0.997	1.032	0.111
Gender	0.998	0.721	1.382	0.991	0.895	0.573	1.399	0.627
Alcohol history documented	1.413	1.020	1.958	**0.038**	1.503	0.984	2.295	0.059
Tobacco smoking history	0.883	0.636	1.225	0.455	0.663	0.429	1.026	0.065
Radiation therapy	0.906	0.649	1.264	0.561	0.648	0.404	1.040	0.073
Neoplasm histologic grade	1.037	0.839	1.282	0.736	1.014	0.747	1.376	0.929
Clinical T	1.151	0.989	1.341	0.070	1.154	0.814	1.637	0.422
Clinical N	1.172	1.003	1.370	**0.046**	1.251	0.952	1.643	0.108
Clinical stage	1.134	0.959	1.339	0.141	0.787	0.496	1.249	0.309
Pathologic stage	1.354	1.128	1.625	**0.001**	1.617	1.188	2.199	**0.002**
Tumor site (sorted by CAV1 expression level)	1.020	0.982	1.059	0.303	1.014	0.884	1.162	0.844
Tumor site (sorted by CAV2 expression level)	1.027	0.985	1.070	0.214	1.013	0.874	1.175	0.861

## Data Availability

The bioinformation data generated and analyzed during the current study are available in the TCGA and CTD repositories [https://portal.gdc.cancer.gov, http://ctdbase.org]. The datasets of clinical information obtained from WHUSS during the current study are not publicly available due privacy of patients but are available from the corresponding author on reasonable request.

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
