# Peer review of "Prognostic Value of CAV1 and CAV2 in Head and Neck Squamous Cell Carcinoma"

_biomolecules, 2023, doi:10.3390/biom13020303_

Round 1

Reviewer 1 Report

Dear Authors,

 My comments on the manuscript entitled ««Prognostic value of CAV1 and CAV2 in head and neck squamous cell carcinoma» Jingyu He, Simin Ouyang, Yilong Zhao , Yuqi Liu , Yaolong Liu , Bing Zhou , Qiwen Man, Bing Liu, Tianfu Wu:

I found this article very interesting to read. The authors use multiple databases for bioinformational search and confirm their hypothesis by conducting experiments.

 However, I have several comments:

1.      Sub-chapter # 2.8 IHC: there is no information on the patients with HNSCC, who were included in the study.

2.      Line 197 :  not a good description of the results, in both cases the correlation was moderate, but for CAV1 it was stronger than for CAV2. Correlation coefficient between 0.4 and 0.6 is considered moderate. (DOI: 10.1213/ANE.0000000000002864, doi: 10.1016/j.tjem.2018.08.001)

3.      Line 228: "HPV statues" misprint (HPV status)

4.      Line 271: the authors compare the level of expression of targets with the size of the primary tumor and the presence of metastases, but not with "multiple clinic pathological", as indicated in the text. Various clinical parameters include primary tumor size, presence of metastases, tumor differentiation (histology), inflammation, keratinization, tumor location (site), etc. Please correct or clarify.

5.      Line 282 : when analyzing survival, did the authors take into account the HPV status of patients?

6.      Line 300: table 2, in my opinion, contains a lot of information. Maybe it's worth listing only significant results? Note that the table title is separate from the table itself.

7.      Discussion section: it is not clear why the authors mention PD1/PDL1 as a "significant biomarker". How does this explain the relevance of the study?  

In general, the study is of great significance. The results obtained add to the knowledge about the involvement of membrane proteins of receptor-independent endocytosis in tumor growth.

Best regards, reviewer

Reviewer 2 Report

The authors have used TCGA data to look at the expression of CAV1 and CAV2 in HNSCC samples compared to normal tissue using a variety to bioinformatics tools. I have two main comments to make:

1. The methods are poorly described, starting with the samples used (n changes throughout the paper - fig 3 seems to indicate n=544?). The validation cohort is not mentioned in the methods, only in the ethics statement at the end of the article - this is a critical omission. In many instances, the authors state what software they have used, but not how it has been used/applied to produce the results. For example, using the CancerSEA database to find the most HNSC-related genes - how did you do this? What (statistical) method or filter was applied? What threshold? etc. What was the sample size? Likewise with c-CBioPortal (2.4) and other places.

2. The sample was split into high & low expression of CAV1 & CAV2 - I presume using the median values in the samples? What were these values? How did this compare the values in the normal samples? A histogram or some other visual would be good here. How robust are the findings to the choice of split value? Include known confounders/prognostic indicators in the analysis, even if not significant in your cohort - you should still include them in the Cox models (good statistical practice).

Additionally, the presentation of the results needs work - many of the figures are not needed as they replicate the same information (e.g. Fig 1a & b). Figure captions should be more informative and it would be good if the text includes mean +/- SD instead of the only summary statistic being the p-value. For many of the boxplots, there is substantial overlap between tumour and normals it seems. 

Consider putting results for CAV1 (or CAV2) in supplementary if similar. Table 2 needs to be condensed, and methods missed the statement that only univariate sig variables were added to multivariate models (?). Was any further model selection considered?

It is unclear what is meant by the sentence "all data are significantly correlated with each other after Welch's t-test (fig 3)".

The analysis has shown only that CAV1 (or CAV2) are significantly associated with outcome not that either of them are predictive - that requires more analysis. Please moderate the language used accordingly.

Reviewer 3 Report

I want to congratulate the authors on a quality piece of research.

First, the article needs English spelling and grammar editing. For example, sentences like in line 34 – which independently implying poor progression. Plurals and singulars in the whole article need to be revised. Also, correction of terms and spelling – disease-free survival DFS, progression-free survival PFS (not progression free interval), and so on.

 The abstract is too long, try to condense it. There is no need for (https://xenabrowser.net/) – these will be stated later in the materials and methods section.

Make figures bigger. Especially 1a, 1c, and 2b lose visibility and clarity. Later too 3+

Part 3.4 try to explain the relationship between CAV mRNAs and HPV, why do HPV-related tumors have a more favorable prognosis?

3.6 statistics and survival analysis is not a suitable title for this section, move table 1 to an earlier point of the text, e.g. results, as this is too far behind all different unrelated data

The whole results section needs reordering-remaking-revising as this for its "too much" information heavy and the reader loses interest.

Round 2

Reviewer 2 Report

The authors have made some of the changes requested in this revision, however I still believe that the statistical analysis requires further work to bring it up to publishable standards. There are inconsistencies in the treatment of prognostic factors in the cox regression analyses that need to be addressed. A set of known prognostic factors (age, sex, smoking status etc) should be included in ALL multivariate analyses regardless of univariate significance. This has not been done. Additionally, using Cox regression, the expression level of the CAV genes can be included as continuous variables. They only need to be dichotomised for visualisation using KM curves. Table 2 does not need to include survival outcomes as they are better analysed (and are subsequently) using KM & Cox regression.

There are too many figures in the article, and not all are useful. We do not need to see the KM curves up to 3yrs and then 5yrs - same information is shown at 5yrs, except slight change in p-value. Concentrate on one survival outcome & move the others to supplementary. It will improve the readability of the paper. Likewise, Fig 1a/b - the information is the same - Fig 1b shows the row for HNSCC from Fig 1a essentially. Remove one of them. Likewise Fig 2 - the boxplots are much clearer than the heatmaps, and show the same information. Other figures also require careful consideration before keeping them.

Minor:

- Please show p-values to a consistent number of decimal places

- Figure captions still need some work (mean/sd not always added in correct places)

Reviewer 3 Report

The changes made to the article are sufficient. When its all in order, the article is suitable for publication.

Author Response

Comment 1 The changes made to the article are sufficient. When it’s all in order, the article is suitable for publication.

Response: Thank you very much for your valuable comments on our article. We have adjusted the content and structure of the article according to the comments. It is our great honor to get your approval.

Round 3

Reviewer 2 Report

Again, while some of the issues raised previously have been addressed, not all have. Please see my previous comments re Table 2 - survival outcomes can, and indeed should, be removed. The methods should clearly state that CAV1/2 expression was continuous for the Cox regression not high vs low, if indeed that was the case. 

Fig 2a remains superfluous and can be removed, unless there is a significant change in sample numbers between 2a and 2b, but this is not stated anywhere.

Additionally, did you consider including site of tumour in the cox regression (given the differences in expression levels by site)? For the IHC cohort, still minimal details are given. Were these the only HNSCC patients in that time period? Age/sex/smoking/ etc for them - if they have a different profile to the TCGA cohort, then it might impact conclusions. Please show this information!
